# Solid-Phase Reference Baths for Fiber-Optic Distributed Sensing

**DOI:** 10.3390/s22114244

**Published:** 2022-06-02

**Authors:** Christoph K. Thomas, Jannis-Michael Huss, Mohammad Abdoli, Tim Huttarsch, Johann Schneider

**Affiliations:** 1Micrometeorology Group, University of Bayreuth, 95447 Bayreuth, Germany; jannis-huss@uni-bayreuth.de (J.-M.H.); mohammad.abdoli@uni-bayreuth.de (M.A.); johann.schneider@uni-bayreuth.de (J.S.); 2Bayreuth Center of Ecology and Environmental Research (BayCEER), University of Bayreuth, 95447 Bayreuth, Germany; 3College of Earth, Ocean, and Atmospheric Sciences, Oregon State University, Corvallis, OR 97331, USA; 4Geosciences Mechanics Shop, University of Bayreuth, 95447 Bayreuth, Germany; tim.huttarsch@uni-bayreuth.de

**Keywords:** distributed temperature sensing, fiber-optics, Raman scattering, atmospheric turbulence, hydrology, calibration, thermoelectric effect, Peltier

## Abstract

Observations from Raman backscatter-based Fiber-Optic Distributed Sensing (FODS) require reference sections of the fiber-optic cable sensor of known temperature to translate the primary measured intensities of Stokes and anti-Stokes photons to the secondary desired temperature signal, which also commonly forms the basis for other derived quantities. Here, we present the design and the results from laboratory and field evaluations of a novel Solid-Phase Bath (SoPhaB) using ultrafine copper instead of the traditional mechanically stirred liquid-phase water bath. This novel type is suitable for all FODS applications in geosciences and industry when high accuracy and precision are needed. The SoPhaB fully encloses the fiber-optic cable which is coiled around the inner core and surrounded by tightly interlocking parts with a total weight of 22 kg. The SoPhaB is thermoelectrically heated and/or cooled using Peltier elements to control the copper body temperature within ±0.04 K using commercially available electronic components. It features two built-in reference platinum wire thermometers which can be connected to the distributed temperature sensing instrument and/or external measurement and logging devices. The SoPhaB is enclosed in an insulated carrying case, which limits the heat loss to or gains from the outside environment and allows for mobile applications. For thermally stationary outside conditions the measured spatial temperature differences across SoPhaB parts touching the fiber-optic cable are <0.05 K even for stark contrasting temperatures of ΔT> 40 K between the SoPhaB’s setpoint and outside conditions. The uniform, stationary known temperature of the SoPhaB allows for substantially shorter sections of the fiber-optic cable sensors of less than <5 bins at spatial measurement resolution to achieve an even much reduced calibration bias and spatiotemporal uncertainty compared to traditional water baths. Field evaluations include deployments in contrasting environments including the Arctic polar night as well as peak summertime conditions to showcase the wide range of the SoPhaB’s applicability.

## 1. Introduction

Fiber-Optic Distributed Sensing (FODS) applications in geo- and marine sciences, engineering, and industry have gained importance over the past decade, and now include the field of meteorology, hydrology, geology, oceanography, and many industrial applications. The spatially explicit nature of the in situ collected observations via installed fiber-optic cables allows for capturing unprecedented spatial scales at fine resolution across several orders of magnitude (from mm to km) at fine temporal resolutions (from s to hours) depending on the geometry of the sensing cable(s) network. The latter may include vertical profiles [1,2,3], horizontal transects [4,5,6], and 2-D [7], quasi, or full 3-D mesh grids [8,9] to name a few. The list of derived environmental quantities in the fields of meteorology and surface hydrology includes temperature, flow velocity and direction, turbulence kinetic energy and thermal dissipation, solar and terrestrial radiation, salinity, air humidity, and soil moisture content to name a few. The reader is referred to Thomas and Selker [10] for a list of applications and techniques in these fields. Fiber-optic point and distributed sensing is also evolving rapidly in the neighboring marine sciences, with observable quantities including temperature, salinity, pressure, chemical and structural parameters, velocity, and turbulence [11,12]. Since the basis for the majority of terrestrial FODS applications is formed by the Distributed Temperature Sensing technique (DTS [13]), the temperature of a reference section of the fiber-optic cable sensor needs to be independently measured and known to translate the raw measurement quantities into temperatures, a process commonly referred to as calibration. This calibration procedure translates the primary light measurement quantities of the intensities of the Raman Stokes and anti-Stokes bands, which are sensed by the DTS instrument, into the desired secondary temperature signal.

Traditionally, FODS applications use volumes of water of tens to hundreds of liters contained in thermally insulated containers to hold the reference fiber-optic cable sections used for online and post-field calibration, which are heated or cooled using aquarium heaters or crushed ice, respectively. This traditional “water bath” is continuously stirred mechanically, or mixed using water pumps or bubblers to prevent stratification and thus undesired temperature differences, which is commonly monitored using at least one platinum-wire thermometer inserted into the liquid. Since the fundamental calibration has three unknowns, at least three reference sections with at least two contrasting temperatures and measurement locations (length along the fiber, LAF) are needed to perform the calibration. This mathematical requirement is commonly met in practice by routing several sections of the fiber-optic cable sensor through reference baths whose temperatures ideally span at least the range of temperatures to be measured, commonly referred to as “cold bath” and “warm bath”. The warm bath is typically heated electrically, while the cold bath is made by periodically adding crushed ice or ice cubes to the water until the bath has reached the ice’s melting point, which results in a two-phase (solid and liquid) mix. The uniformity and stationarity of these reference baths are crucial to achieving a high calibration quality by reducing measurement uncertainty both across the reference sections and the actual fiber-optic sensing array for the intended geoscientific or industrial applications. In short, the better these baths perform, the more meaningful are the collected observations.

Much attention has been given to enhancing FODS measurement quality by improving the light emitting and sensing properties of the DTS instruments and solving the mathematical calibration procedure [14,15,16], but to date little effort has been devoted to improving the physical reference environment, i.e., the reference baths. While, in theory, a well-mixed liquid-phase water bath offers an easy to construct, inexpensive, mobile, and environmentally friendly temperature reference, in practice, they suffer from several significant shortcomings increasing the calibration and thus measurement uncertainty which may deteriorate the signal quality significantly depending on the type of application. Shortcomings include (1) instationary bath temperature due to (a) imperfect insulation from outside conditions often subject to external, e.g., through radiative forcing, heating and cooling, and (b) heating and cooling cycles of the electronic control circuit or addition of crushed ice to maintain the bath’s temperature; (2) systematic spatial heterogeneity of the bath temperature because of imperfect mixing particularly at the ice–water interface and the corners of the water volume; (3) limited range of environmental conditions excluding subfreezing environments unless liquids with a lower freezing point are used; (4) the need for large quantities of water or liquid to curb bath instationarity by increasing the heat storage capacity; (5) significant maintenance needs when periodically refilling the crushed ice in the cold bath for longer-term deployments beyond a few hours; (6) the potential growth of algae and other flora/fauna particularly in the warm baths without the use of biocides; (7) risk of electric shock for leaky containers and during bath maintenance when resistively heated stainless-steel sheated cables are used for actively heated FODS applications. Hence, we identify the need to overcome the shortcomings of these traditional water baths with the goal of improving the calibration and signal quality by reducing its uncertainty through the use of solid-phase baths, we here term “SoPhaB”, as a universal temperature reference for FODS applications. Their design is intended to provide a precise and accurate, easy-to-maintain reference with a much extended range of bath temperatures.

In Section 2 we present the SoPhaB design, compute its fundamental thermal properties, and describe their use, as well as provide a brief description of the laboratory and field conditions used for evaluation. Section 3 contains the evaluation results across contrasting environmental conditions, which are discussed and compared to selected applications of traditional water baths in Section 4. In Section 5, we summarize SoPhaB performance, give practical advice for their optimal performance, and highlight potential areas for future improvements. Appendix A contains lists of components and resources needed for constructing SoPhaBs.

## 2. Materials and Methods

The SoPhaBs described here have an improved design compared to an earlier type described in Lapo et al. [17] and Zeller et al. [18], particularly with respect to the thermoelectric control, carrying case, and handling and routing of the fiber-optic cables.

### 2.1. SoPhaB Design and Use

One SoPhaB consists of six tightly interlocking parts (Figure 1), which create a reference volume of uniform temperature completely enclosing the fiber-optic cable section coiled around the internal groove cut into the inner core (mass, *m* = 8.2 kg) of 6.25 mm depth and 35 mm width (Figure 2c). The circumference of the inner groove is ≈0.50 m, which is important to know when coiling the fiber-optic cable during deployment to create a reference section of sufficient length. The groove containing the fiber-optic cable is completely surrounded by an outer ring (*m* = 3.1 kg) of 12.5 mm thick walls, and contained by a top (*m* = 5.2 kg) and bottom (*m* = 5.3 kg) slab of 20 mm thickness. Two small pins ensure that parts are properly centered which is important for uniform heat conduction across the SophaB and helps during assemblage. The fiber-optic cables enter and leave the SoPhaB through two slits of 5 mm width and 30 mm length cut into the outer ring (Figure 2d). Two shallow grooves (4 mm wide, 2 mm deep), one located in the outer wall of the inner core (Figure 2c) and one in the inner wall of the outer ring, contain the reference platinum-wire thermometers (Pt100) permanently attached by a heat conductive adhesive to measure its body temperature directly next to the fiber-optic cables (Figure 2c). Their electrical wires leave the SoPhaB through the same slits used for routing the fiber-optic cables. The total outside dimensions are 197 mm in diameter and 85 mm in height with a total mass of 21.8 kg (Figure 2d), which considering the density, ρ, of ultrafine copper (99.99%, at 20 °C) of 8930 kg m${}^{-3}$ corresponds to a volume of $V_{\mathrm{copper}}$ = 2440 cm${}^{3}$. The volume of the internal groove containing the fiber-optic cables is $V_{\mathrm{groove}}=V_{\mathrm{SoPhaB}}-V_{\mathrm{copper}}=8.5$ cm $\cdot \pi {\left(\frac{19.7cm}{2}\right)}^{2}-V_{\mathrm{copper}}\approx$ 150 cm${}^{3}$, which is relevant for how much cable can be contained in the SoPhaB.

The heat transfer to or from the SoPhaB to change its heat content is analogous to charging and discharging a capacitor with a time constant given by
(1)$$\tau =R\cdot C=\left(\frac{1}{l\cdot \lambda }\right)\cdot \left(\rho cAd\right)=\left(\frac{d}{A\cdot \lambda }\right)\cdot \left(\rho cAd\right)=\frac{1}{K}d^{2},$$
where *R* is thermal resistance (K W${}^{-1}$), *C* the capacitance (J K${}^{-1}$), *l* some characteristic dimension (m), *A* the surface area, *d* the distance over which the heat is transported, λ the heat conductivity (W m${}^{-1}$ K${}^{-1}$), *c* the specific heat (J kg${}^{-1}$ K${}^{-1}$), and *K* the heat diffusivity given by $K=\frac{\lambda }{\rho c}$ (m${}^{2}$ s${}^{-1}$). To heat or cool the copper body, Peltier elements are attached to the top slab, which contains a shallow groove for the NTC thermocouple attached to the controlling electronics for sensing the body temperature (Figure 2b). Tests with Peltier elements attached to both the top and bottom slab did not yield better bath performance (not shown here). Since the speed of the heat transfer through the copper body is of utmost importance for its performance, we can estimate an effective response time to the thermoelectric regulation by estimating the lower and upper boundaries: the minimum distance from the outer edge of the Peltier element attached to only one side of the copper body to the center of the inner groove holding the fiber-optic cable is $d_{min}$ = 82 mm, while the maximum distance between the top’s center and the outer edge of the bottom is $d_{max}$ = 130 mm. Setting *K* = 1.14 × 10${}^{-4}$ m${}^{2}$ s${}^{-1}$, Equation (Equation 1) evaluates to $\tau_{min}$ = 59 and $\tau_{max}$ = 148 s, respectively. These times assume the heat pulse travels uniformly across the entire body, which will depend upon how tightly the interlocking parts fit together to not create gaps hindering heat transport. The actual response time of the SoPhaB and that of the fiber-optic cable coiled around its inner core will be somewhere in between these boundaries. The response time of the fiber-optic cable (see Table 1) used in most deployments selected for this study was calculated by Thomas et al. [7] and evaluated to ≈1.2 s, and thus adds little to the overall temporal response. The validity of Equation (Equation 1) to predict a meaningful time response was confirmed by a series of laboratory tests using slabs of ultrafine copper of varying thickness. The experimental findings agreed with their predictions within seconds (not shown here). For an optimal bath performance even during the initial strongly instationary phase until the SoPhaB has reached its setpoint starting from the initial ambient temperature, the heating or cooling rate shall be small compared to the computed response time to avoid systematic temperature differences deteriorating calibration performance. Recall that one reference thermometer Pt100 is in direct physical contact with the fiber-optic cable coiled around the inner core, and the other is embedded in the inner wall of the outer ring and thus within a few mm of the fiber-optic cable and subject to enhanced longwave radiative transfer due to the applied paint described below. Hence, the time needed for a heat pulse to travel between these locations is negligible compared to the response time from the Peltier element to the fiber-optic cable section. The energy needed to change the SoPhaB’s body temperature is 8.4 kJ K${}^{-1}$, which is useful to know for properly dimensioning the Peltier elements. Since cooling is less efficient than heating with Peltier elements [19], the cold SoPhaB uses a stack of two identical elements (Figure 2b), while the warm SoPhaB only uses one to achieve similar rates of temperature change.

The heat of the SoPhaB propagates through its body by molecular diffusion, here termed conduction, until it reaches the walls of the ring and the inner core which are in direct contact with the fiber-optic cable to be referenced. Assuming an absence of turbulent diffusion, here termed convection, in the cut-out groove of the inner core, the dominant heat transport mechanism to the fiber, which may or may not physically touch the inner core, is radiative transfer due to conduction in air being much smaller in comparison. Note that the response time and spatiotemporal variability of the readings from the fiber-optic cable wound around the inner core is independent of direct physical contact, as it is dominated by the time scale given in Equation (Equation 1). Tightly wrapping the fiber-optic cable around the inner core helps slide the outer ring into place when closing the SoPhaB. However, applying too much tension will induce unwanted strain and result in changes in the optical light conducting properties, which may lead to step changes in the differential attenuation of the Raman intensities. Polished metal surfaces have a relatively small emissivity ϵ≈ 0.4 hindering an efficient radiative heat transfer—we recall that $I=\epsilon \cdot \sigma_{SB}\cdot T^{4}$, with *I* being the radiative flux density and σ the Stefan-Boltzmann constant. Hence several layers of high-emissivity paint (HERP-LT-MWIR-BK-11, paints.labir.eu) with ϵ = 0.95 were applied to the inner wall of the outer ring and the outer wall of the inner core to enhance heat transfer. The outside jackets of the fiber-optic cables we used for evaluation are made from polyvinyl chloride or polyethylen with emissivities similar to that of the paint.

When the copper body is installed in its carrying case, an aluminum block connects the outer side of the Peltier elements to the electrically aspirated CPU cooler which acts as a heat sink or source depending on whether the SoPhaB is cooled or warmed (Figure 2a,d). An important step is to apply a thin coat of heat conducting paste (λ = 0.9 W m${}^{-1}$ K${}^{-1}$, heat-conductive metal oxides in silicon oil, Felder GMBH Loettechnik, Germany) to all surfaces touching each other including the copper parts, Peltier elements, aluminum block, and CPU cooler to ensure optimal heat conduction by bridging the small, but finite gaps due to machining precision and mechanical needs during assemblage. Earlier attempts using a heat conducting foil resulted in much larger response times and spatial temperature differences across the copper core increased by a factor of ≈10 compared to the heat conducting paste. All resources needed to construct one pair of SoPhaBs are listed in Table A1 and Table A2 in Appendix A.

### 2.2. Field and Laboratory Evaluations

We selected three case studies for evaluating the performance of the SoPhaBs in contrasting temperature environments, and compare our findings to two FODS deployments using traditional liquid-phase water baths in similarly contrasting conditions. Each selected experimental FODS dataset contains observations over 300 min but at different temporal and spatial resolutions (Table 1). All experiments with the exception of the intentional instationary “Windtunnel” data represent constant-temperature field evaluations, which are discussed in depth in Section 3 and Section 4. All fiber-optic data were calibrated using the mathematical procedures described by des Tombe et al. [16] implemented in the open-source calibration software pyfocs [20] except for “Windtunnel”, for which the manufacturer’s internal calibration routines were used (Table 1). All experiments used single-core multimode fiber of 50 μm inner diameter. We focus solely on the fiber-optic cable sections contained in the reference baths, with readings from the other fiber sections containing the actual measurement not being reported here. Since FODS observations are explicit in both space and time domains, we apply the common Reynold’s decomposition for their respective instantaneous readings, perturbations, and averages using the following symbols:(2)$$\mathrm{Space}:x=\hat{x}+\langle x\rangle$$
(3)$$\mathrm{Time}:x=x^{\prime }+\bar{x}$$

The symbols on the left-hand side of the above equations are the instantaneous readings of any quantity *x*, which in our case are either the fiber-optic temperatures $T_{f}$, or the ratio of intensities $r_{I}$ (see the start of Section 3 for their definitions). The $\hat{\;}$ and ${}^{\prime }$ symbols represent the excursions from their spatial and temporal averages, which are denoted by the angle brackets and overbar, respectively.

For the “Windtunnel” evaluation, one SoPhaB containing a section of approximately 5 m length toward the end of the described fiber-optic cable was placed in the wind tunnel of about 21 °C with the CPU heat sink element attached, but electric fan detached. The ventilation speed of the CPU heat sink was set through the wind tunnel controls (for details of the tunnel see [21,22]). The setpoint for the two Peltier elements was initially set to cool to 19.0 °C for the first 30 min, then adjusted to 7.2 °C for the remainder of the experiment. In addition to the two standard reference thermometers built into each SoPhaB (rings’ inner wall, core’s outer wall), one additional thermometer of identical make was attached to the outer edge of the bottom slab using heat conductive paste at the maximum distance from the Peltier elements attached to the center of the top slab. The SoPhaB was completely enclosed in insulation material made from polyurethane foam with a minimum thickness of 40 mm in all directions with the exception of the cutout for attaching the external CPU heat sink to the Peltier elements via the aluminum block. The setup was almost identical to the SoPhaB contained in the carrying case for the selected field evaluations.

The “Arctic polarnight” dataset was collected in the Adventdalen valley near Longyearbyen, Svalbard (78.202° N, 15.826° E) during the polar night in February 2022. The reference baths contained sections of about 1.5 and 4.5 m lengths in the warm and cold SoPhaB, respectively located at the beginning of the cable at the indicated LAFs. The remainder of the fiber-optic cable was arranged in a high-resolution coil-wrapped profile across the snow-air interface of 1 m height (similar to [23]). The SoPhaBs were housed inside the old Aurora station at a constant room temperature of approximately 6 °C, with outside minimum temperatures of around −24 °C for the selected case.

The “Urban summer” dataset was collected in August 2021 in the city of Münster, Germany, (51.947° N, 7.641° E) on the hottest day during the summer campaign. The reference baths selected for evaluation contained about 3.5 m fiber-optic cable in each SoPhaB (see Figure 4b) located at the beginning of the cable at the indicated LAFs. The remainder of the fiber-optic cable was arranged in a high-resolution coil-wrapped profile extending from the concrete ground into the air across a total height of 3.8 m. The SoPhaBs were housed inside an air-conditioned field trailer, whose inside temperature showed a dampened response to the intense solar forcing during the selected clear-sky morning. Outside temperatures ranged between 15 and 26 °C, while the air inside the container ranged between 15 and 18 °C.

For comparison with the SoPhaBs, we selected observations from the “Arctic pier” deployment in March 2020 in the town of Ny-Ålesund (78.928° N, 11.922° E) which used liquid-phase baths filled with a mixture of water and propylene glycol (antifreeze) for the deployment in the subzero Arctic conditions. Both the cold and warm baths contained pumps and aquarium heaters for temperature control, with the addition of crushed ice to the cold bath at the lowest possible setpoint to stabilize its temperature at about −2.5 °C. Each bath contained an approximately 7 m long section of the fiber-optic cable at its beginning (see Figure 4c). The remainder of the fiber-optic cable was arranged in a high-resolution coil-wrapped profile sampling temperatures across the sea ice-air interface of approximately 3 m length. The baths were contained in insulated party coolers of approximately 80 L with 60 mm thick walls, located in the open on the old pier protruding into the fjord. Outside air temperatures ranged between −13 and −17 °C.

The second dataset “Grass summer” selected for comparison is from Thomas et al. [7], collected at the Botany and Plant Pathology lab of Oregon State University, Corvallis, OR, USA (44.567° N, 123.242° W) above short grass. The fiber-optic cable sections contained in the liquid-phase water baths were approximately 5.5 and 3.5 m long for the cold and warm baths, respectively (see Figure 4d). Crushed ice was added to the cold bath to keep its temperature nearly constant at 0 °C (see Figure 4h), while the temperature of the warm bath was not actively controlled but followed the diurnal temperature amplitude with some phase delay and at a dampened amplitude. Both baths used aquarium bubblers to enhance mixing. The water baths were located in open shade with outside temperatures of about 10 °C during the selected early morning period. The remainder of the fiber-optic cable was arranged in a two-dimensional evenly spaced array of 0.25 m grid spacing of approximately 8 m × 8 m size.

## 3. Results

The basic quantities measured by the DTS instrument employing the inelastic Raman backscatter of photons are the intensities of the red-shifted Stokes ($I_{s}$) and blue-shifted anti-Stokes ($I_{as}$). These quantities are unaffected by any subsequent mathematical calibration method. Since the user-desired secondary quantity, the fiber-optic cable temperature $T_{f}$, is proportional the logarithm of their ratio, $T_{f}\propto log\left(\frac{I_{as}}{I_{s}}\right)=r_{I}$, we will use statistical measures of the right-hand-side quantity simply referred to as “the ratio of intensities” to evaluate the reference bath performance as a first fundamental step similar to Lapo and Freundorfer [20], before we proceed to comparing it against the reference temperatures from the platinum-wire thermometers $T_{Pt100}$.

### 3.1. Windtunnel Evaluation

Observations in the wind tunnel yielded a uniform distribution of the ratio of intensities across the entire reference section of the fiber-optic cable (Figure 3a). Its temporally averaged spatial perturbation over the 300 min measurement period was $\bar{\hat{r_{I}}}<$ 5 × $10^{-4}$ showing a minor spatial linear trend, with its twofold temporal standard deviation 2${\left(\bar{{\hat{r_{I}}}^{2}}\right)}^{0.5}\leq$ 1 ×$10^{-3}$. A plausible explanation for the observed linear trend results from the experimental setup and is discussed in the subsequent section. The approximate finite differences between adjacent LAF-bins $\Delta r_{I}$ was very evenly distributed across the reference section with minor deviations located at the start of the reference section with $|\bar{r_{I}}|<$ 2 ×$10^{-4}$ (Figure 3b). The systematic trend observed in the temporal mean of the ratio of intensities was not evident in this quantity. The agreement between spatially averaged fiber-optic $\langle T_{f}\rangle$ and reference $\langle T_{Pt100}\rangle$ temperatures, computed over all three thermometers, during this intentional strongly instationary phase was remarkable (Figure 3c) with only minor systematic differences <0.1 K in the first 90 min, which will be explained later. During the cooling phases, the spatial differences across the SoPhaB’s copper core were <0.10 K between the outer wall of the inner core and the edge of the bottom slab, and <0.04 K between its parts in direct proximity to the reference section of the fiber-optic cable (Figure 3d). In contrast, these differences were negligible during the stationary phase close to room temperature, while they persisted when the SoPhaB reached its setpoint at ≈14 K below room temperature.

The effective temperature change of the SoPhaB achieved by the stack of two Peltier elements working at the maximum electrical power (measured 65 W, nominal 280 W total) was 13 K over 240 min, or an average cooling rate of 0.055 K min${}^{-1}$ equivalent of 455 J min${}^{-1}$ when recalling its previously computed energy demand of 8.4 kJ K${}^{-1}$. The effective efficiency of the Peltier cooling was thus approximately 12% and 3% compared to its measured and nominal electrical power, respectively, at the ventilation speed of ≈8 ms${}^{-1}$ in the wind tunnel.

### 3.2. Field Deployments

For the selected field experiments the SoPhaBs were at their setpoint temperatures in a controlled dynamically stationary state. The temporal mean and variability of spatial perturbations of the ratio of intensities $\hat{r_{I}}$ were much more homogeneous and smaller with <1 × $10^{-4}$ and <5 × $10^{-4}$, respectively, across the fiber-optic cable reference sections in both the warm and cold baths, compared to “Windtunnel” (Figure 4a,b). Even the much shorter warm SoPhaB during the “Arctic polarnight” dataset consisting of only four bins at the measurement resolution of 0.254 m did not have increased spatial and temporal variability or trends. For “Urban summer”, the cold SoPhaB showed a minor spatial linear trend across the 3.5 m long reference section, which was similar in magnitude to that of “Windtunnel”. The spatial average $\langle T_{f}\rangle$ and standard deviation ${\langle T_{f}\prime^{2}\rangle }^{0.5}$ of the fiber-optic temperatures compared well to the readings of the reference (Pt100) thermometer located on the outer wall of the inner core next to the fiber-optic cable (Figure 4e,f and Figure 5a,b). The interquartile range (IQR) of instantaneous spatial deviations between fiber-optic and reference Pt100 temperatures $\left(\hat{T_{f}-T_{Pt100}}\right)$ across the entire reference section spanned between 0.07 and 0.10 K for all observations in the SoPhaBs. It was evenly distributed across negative and positive values with a maximum bias of −0.01 K in the case of the warm SoPhaB for “Urban Summer” (Figure 5e,f), where the bias is defined as the median of the probability density function (pdf). For this quantity, we use distribution-free rather than Gaussian statistics since the pdfs are non-Gaussian, particularly in the case of the water baths.

In contrast, analyzing the ratio of intensities for the water baths revealed distinct systematic spatial patterns with maximum changes on the order of $\bar{\hat{r_{I}}}$≈ 4$\times 10^{-3}$ in the case of the warm bath of “Arctic pier”, but a temporal variability of surprisingly similar magnitude compared to the SoPhaBs (Figure 4c). The much larger band of variability given by 2${\left(\bar{{\hat{r_{I}}}^{2}}\right)}^{0.5}\approx$ 4 ×$10^{-3}$ also evident in the spatial variability between fiber-optic and reference (Pt100) temperatures (Figure 5h) found for “Grass summer” resulted from the short sampling interval of 1 s, which drastically increases the noise floor of the high-resolution DTS machines (Figure 4d). The temporal course of the spatial average and standard deviation of the fiber-optic temperatures compared well to that of the reference thermometers (Figure 4g,h and Figure 5c,d), but at a significantly larger IQR for the instantaneous spatial deviations ranging between 0.3 and 0.55 K (Figure 5g,h). The magnitude of their biases was ≤0.03 K. For “Grass summer”, an inspection of the time series of temperature deviations between $\langle T_{f}\rangle$ and the reference thermometer revealed a much larger temporal variability for the warm water bath compared to the cold bath (Figure 5d), which, in contrast, did not affect the pdf of the instantaneous deviations in any significant fashion (Figure 5h). The effects of temporal and spatial averaging will be discussed in the next section. The pdfs of the instantaneous deviations were bimodal and strongly non-Gaussian for “Arctic pier”.

## 4. Discussion

The minor spatial linear trend observed in the ratio of intensities for the single SoPhaB used in “Windtunnel” resulted from the use of only one reference section. In such a case, the location-specific differential attenuation in units of dB km${}^{-1}$ for each fiber-optic measurement bin needs to be estimated visually by inspecting the space series and set manually during sampling, while a full post-experiment calibration requires at least three independent reference sections of known temperature [14]. Here, we set it to 0.265 dB km${}^{-1}$ which appeared to have been slightly too small. However, this systematic user error did not degrade the quality of the SoPhaB performance. Overall, the SoPhaBs provide an excellent spatially homogeneous thermal reference environment in spite of the strongly instationary cooling phase. The maximum achieved cooling rate of 0.05 K min${}^{-1}$ closely matched the IQR of the instantaneous differences between the reference thermometers and $T_{f}$ across the SoPhaB < 0.05 K when recalling the estimated effective response time of about 1 min. Stronger cooling rates may thus lead to greater spatial temperature differences during instationary phases and are not recommended given the SoPhaB’s current dimensions. Hence, the low efficiency of the Peltier elements of ≈12% is conducive to the objective of creating a uniform thermal reference environment for the FODS observations. The determined efficiency falls within the typical range of thermoelectric cooling between 5 and 15% [19]. The finite temperature difference of ≈0.07 K measured between the reference thermometer next to the fiber-optic cable and the outer edge of the bottom slab at the lower setpoint of 7.2 °C resulted from imperfect insulation, as heat was obviously leaking into the bottom of the SoPhaB creating a positive difference. We recall, however, that the entire SoPhaB including its insulation was exposed to the high wind tunnel speeds which significantly enhances the heat transport from the ≈14 K warmer environment to the much cooler SoPhaB core through shear-induced convection. This use case is unlike that in field deployments where only the CPU fan is used for ventilating the heat sink while not affecting the carrying case’s temperature. Note that spatial differences across reference thermometers were negligible for the stationary phase at the higher setpoint of 19.0 °C. As mentioned earlier, the use of the heat-conductive paste to connect all parts participating in the transfer is crucial to achieving the desired thermally homogeneous reference environment. Both testing without it and using heat-conductive foil resulted in temperature differences >0.2 K across the SoPhaB core during the instationary phases (not shown here). However, one disadvantage of the heat-conductive paste is that it acts as a very strong adhesive at setpoints below freezing (Figure 4e), making it impossible to reopen the SoPhaB to e.g., inserting or removing fiber-optic cables. For this reason, we chose the Peltier controller of the cold bath to be able to actively heat or cool, so the user can raise the bath’s temperature to ≥10 °C before cable reconfiguration at which point the paste becomes fluid enough to separate its tightly interlocking parts.

We cannot offer a plausible explanation for the slightly enhanced differences between the cold SoPhaB and the reference thermometers compared to the warm bath for “Arctic polarnight” (Figure 5a). We suspect some heat leaking into the SoPhaB core despite the thick insulation of the carrying case driven by the temperature difference of ≈24 K, which may have led to stronger heating and cooling cycles. However, these did not affect the pdfs of the instantaneous differences noticeably or deteriorate the quality of the SoPhaB, particularly in comparison with those from the water baths, which were between 3 and 8 times larger.

We have no explanation for the small spatial trend found in the ratio of intensities found in the cold SoPhaB of “Urban summer”. The calibration was performed double-endedly with a negligible bias < 0.01 K and a small IQR ≈0.05 K. Improper insulation of the carrying case in combination with a directional external heat source conceivably resulting from the warmer trailer heated by solar radiation would have caused a wavy pattern as found in the imperfectly mixed water baths for “Arctic pier”. The small temporal trend of the SoPhaBs’ temperature of ≈0.1 K over the 300 min despite the fixed setpoint suggests that either the heat transfer between the CPU heat sink and the outside environment or the change in the temperature of the electronics affected its ability to regulate bath temperature to some small degree. For optimal performance, the SoPhaBs shall thus be housed in an isothermal environment. However, a temporal trend <0.05 K min${}^{-1}$ estimated from “Windtunnel” does not impact the precision and accuracy of the calibration.

The spatial pattern found in the ratio of intensities in the water baths was caused by the imperfect mixing of the fluid despite the mechanical mixing. The slush mix of crushed ice and antifreeze, as well as the use of the aquarium heating, caused large systematic warmer and cooler pockets of fluid. Since the fiber-optic cable was arranged in a figure eight around a mesh-based support structure to avoid twisting and entanglement of the cable, the pattern consisted of ripples. While the use of sufficiently long fiber-optic reference sections allowed for compensating the temporal variability of the reference section LAF-bins by means of spatial averaging as demonstrated by the small standard deviation similar to that in the SoPhaBs, the instationary fluid’s temperature resulting from the heating cycles in combination with the imperfect mixing caused a much larger IQR and thus poorer precision of the calibration and non-Gaussian statistics. This finding is relevant for all FODS studies seeking to collect averages of observations along profiles and transects requiring high accuracy over time to not induce artifacts by the slowly changing thermal structure of the reference baths. In contrast, studies investigating fast-changing processes such as turbulence in fluids do not suffer from systematic temperature differences caused by imperfect mixing in water baths if their volumes are sufficiently large. In this case, their response to any force is slow compared to the fast-changing process scales as the turbulent signal is typically extracted as the perturbation from a temporal or spatial mean. In comparison though, SoPhaBs offer a much reduced uncertainty (bias, IQR) already at much shorter fiber sections of only several bins at measurement resolution. One needs to consider that the mix of liquid and solid phases in combination with large bath volumes to dampen temperature instationarity and the freezing environment constantly leading to heat loss arguably presents a worst-case scenario for using such reference baths.

The greater temporal standard deviation of the ratio of intensities observed in both reference water baths of “Grass summer” results from the lower signal to noise ratio of the high-resolution DTS instrument, which was also found in Peltola et al. [2], Thomas et al. [7], and Lapo et al. [17]. Even if one compensates for the larger random error arising from the short averaging interval of 1 s compared to all other field evaluations (Table 1) by dividing the reported standard deviation by $\sqrt{10}$, the standard deviation of the ratio of intensities is still twice as large compared to that of the SoPhaBs. This finding is somewhat surprising as the DTS instrument variant with a maximum measurement distance of 2 km was deployed in “Grass summer”, which typically has a much lower noise floor compared to the 5 km variant, used in “Windtunnel”. In the case of DTS instrument samples at a high temporal resolution to observe fast-changing geophysical processes such as turbulence in fluids or the air, much longer reference sections are needed to compensate for the large temporal variability by spatial averaging. In both water baths, some systematic temperature differences were evident, but of much smaller magnitude compared to “Arctic pier”.

Only a few studies report a detailed evaluation of the performance of the reference baths. des Tombe et al. [16] reported spatial differences of up to ≈0.6 K between calibrated FODS and the reference thermometers. Systematic differences across fiber-optic temperatures were ≈1 K for an effective average of 4 s from combining two 2 s averaged channels in a double-ended calibration and warm and cold stirred water baths. Their reference baths consisted of 38 bins at 0.254 m length, which is much longer compared to any of our deployments. The mean biases over a 24 h observational period between their fiber-optic temperatures spatially averaged across the reference section and the reference thermometers were typically ≤0.02 K, with their “warm waterbath 2” showing a distinct spatial trend possibly due to insufficient mixing. These results are in good agreement with our findings for water baths. Their temporal and spatial standard deviations of the instantaneous differences between fiber-optic temperatures and reference thermometers were typically ≈0.14 K, which corresponds well to our IQR of about 0.3 K when considering the differences in statistical measures used. In contrast to our findings in “Arctic pier”, their deviations closely followed a normal distribution allowing for use of Gaussian statistical measures. Hausner et al. [14] reported results for their FODS calibrations using >40 m long fiber-optic reference sections in traditional water baths at ambient and elevated temperatures averaged over 30 s. Their mean root square error and bias between calibrated fiber-optic and reference thermometers were also on the order of 0.2 K and 1 × 10${}^{-4}$, respectively, for the algorithm 1 in [14], which is closest to our calibration method. It is difficult, however, to gauge the between-fiber-optic bin deviations in their reference baths from figures, and ratios of intensities were not reported in the tables. Lapo et al. [17] reported much larger effective biases of ≈0.2 K and larger spatial standard deviations of ≈0.1 K between calibrated fiber-optic temperatures and reference thermometers in the reference sections using an earlier SoPhaB version in a field experiment conducted over several weeks. Their biases showed a distinct diurnal cycle of small magnitude (<0.1 K) due to changes in the temperatures of the trailer, in which the SoPhaBs were housed. The poorer calibration performance statistics are likely a result of much longer and spliced fiber-optic cables resulting in step losses, and remaining unconstrained uncertainty in their single-ended calibration routine rather than a result of spatial temperature differences across the SoPhaBs’ cores. The stationarity of their SoPhaB temperatures over weeks was very good outside of periods affected by power outages and the resulting failure of active bath regulation. Lapo et al. [17] also found much larger spatial and temporal variability in fiber-optic reference bath temperatures using the identical 5 km variant of the high-resolution compared to the ruggedized slower-response DTS machines, which could also not solely be explained by the increased spatial and temporal integration of the latter.

## 5. Conclusions

We demonstrated that the use of solid-phase baths, here termed SoPhaBs, for referencing fiber-optic distributed sensing measurements offers many advantages over traditional liquid-phased baths. Collectively these advantages allow for much shorter reference sections of fiber-optic cables of only several bins at measurement resolution, at an improved spatial homogeneity of the reference bath ≤0.05 K, excellent stationarity at scales of 0.05 K min${}^{-1}$, and much reduced maintenance needs for long-term installations of weeks and months. Temperature setpoints well below freezing make their use ideal for deployments in cold environments such as Polar regions or industrial applications. In such cases, the cold SoPhaB needs to be actively heated up above freezing for reconfiguration to increase the fluidity of the heat conductive paste connecting all heat-transferring parts. While SoPhaBs can maintain setpoint temperatures different from their environment by several tens of Kelvin with negligible spatiotemporal variability of the copper reference bath allowing for extremely low calibration uncertainty with a bias <0.02 K, the optimal deployment environment offers small temperature changes. From our point of view, these advantages outweigh potential disadvantages such as higher expenses for production and shipping due to the increased weight. While the current SoPhaB’s dimensions should offer sufficient space for several fiber-optic cables to be deployed at sufficient lengths at once, its dimensions can easily be adjusted when fiber-optic cables of larger diameters are used. Further improvements shall include more efficient insulation of the carrying case particularly around the slits contained in the outer ring for routing the fiber-optic cables to limit the undesired heat loss to or gain from the environment, and easier handling of fiber-optic cables when coiling them around the inner core and enclosing them with the outer ring. As general advice, the quality of the reference baths shall receive greater attention during calibration and be included when assessing and reporting the quality of FODS measurements in any application in order to separate measurement artifacts from the true signal of interest.

## Figures and Tables

**Figure 1 sensors-22-04244-f001:**
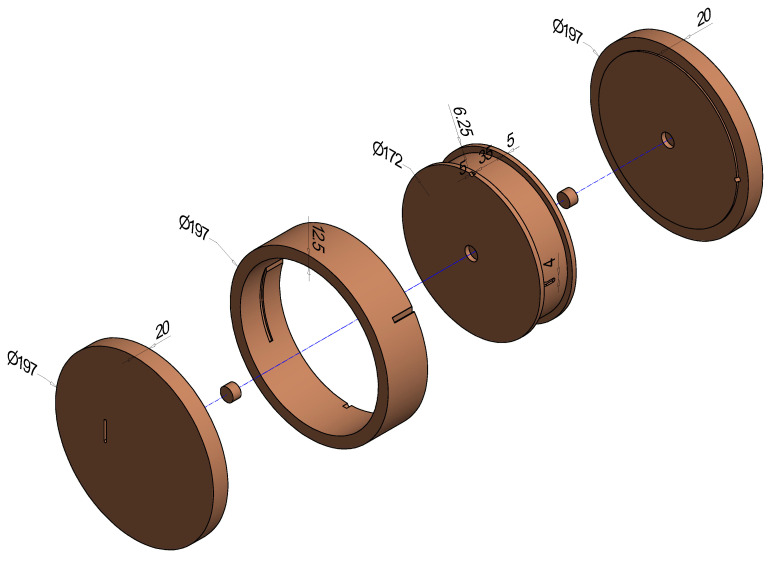
Exploded view of the Solid-Phase Bath (SoPhaB) made from ultrafine copper intended as temperature reference for FODS applications. Its central part, consisting of the inner core and the outer ring, is made from one solid piece of copper by means of electrical discharge machining to create tightly interlocking parts. The total weight of one SoPhaB is approximately 22 kg. All parts and supply materials to construct one pair are listed in Table A1 and Table A2.

**Figure 2 sensors-22-04244-f002:**
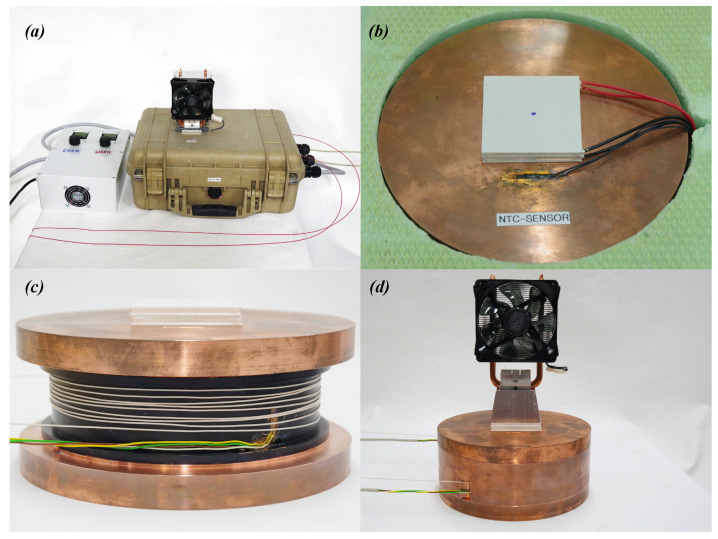
Overview and details of the copper Solid-Phase Bath (SoPhaB) used for referencing fiber-optic distributed mesurements: (**a**) One SoPhaB mounted in carrying case and control electronics as used in the field deployments with a 1.12 mm outer diameter stainless steel, red polyethylen coated single-core multimode fiber-optic cable used in data set “Windtunnel”, see Table 1; (**b**) Detailed view of the top slab with a stack of two Peltier elements of 62 mm × 62 mm dimensions, the NTC thermometer senses the copper body temperature and is connected to the control electronics; (**c**) Detailed view of the inner core painted with high-emissivity paint containing 8 coils of the tightly buffered, aramid reinforced 0.9 mm outer diameter PVC encased single-core multimode fiber-optic cable used in the field experiments (Table 1), also shown is the reference platinum wire (Pt100) thermometer embedded in the outer wall of the inner core connected to the DTS instrument to sense the reference temperature; (**d**) View of the SoPhaB’s assembled copper elements outside of the carrying case with aluminum block and CPU cooler. All parts and supply materials to construct one pair are listed in Table A1 and Table A2.

**Figure 3 sensors-22-04244-f003:**
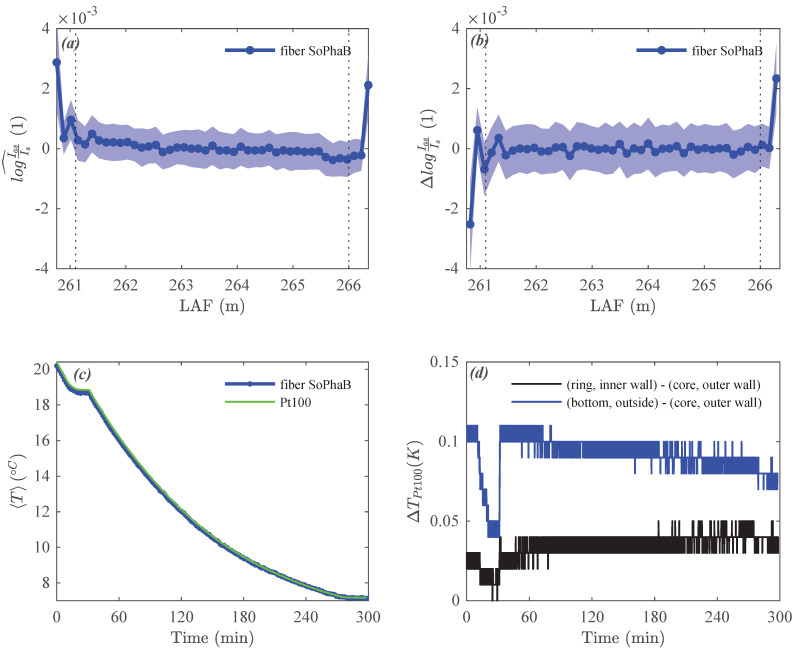
Solid-Phase Bath (SoPhaB) performance from the cooling “Windtunnel” evaluation: Temporal average (solid line with markers) and ±standard deviation (band) of the (**a**) spatial perturbation of the ratio of Raman backscatter intensities, and (**b**) their approximate finite differences between adjacent fiber-optic measurement bins. Time series of the (**c**) spatially averaged fiber-optic and reference (Pt100) temperatures, and (**d**) spatial reference (Pt100) temperature differences across several locations at measurement resolution of 0.01 K. The vertical dotted lines mark the beginning and end of the reference section over which the spatial average was computed.

**Figure 4 sensors-22-04244-f004:**
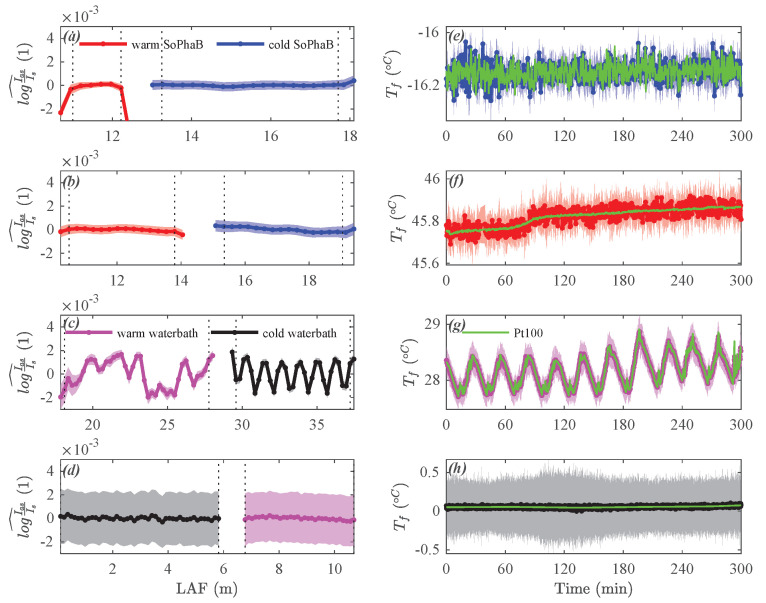
Solid-Phase Bath (SoPhaB) and waterbath performance from the constant-temperature field evaluations: Temporal average (solid line with markers) and ±standard deviation (band) of the spatial perturbation of the ratio of Raman backscatter intensities across the fiber-optic cable reference sections (**a**–**d**), and time series of the spatially averaged (solid line with markers) ± spatial standard deviation (band) fiber-optic and reference (Pt100) temperatures (**e**–**h**) for (**a**,**e**) Arctic polarnight, (**b**,**f**) Urban summer, (**c**,**g**) Arctic pier, and (**d**,**h**) Grass summer. The vertical dotted lines mark the beginning and end of the reference section over which the spatial average was computed.

**Figure 5 sensors-22-04244-f005:**
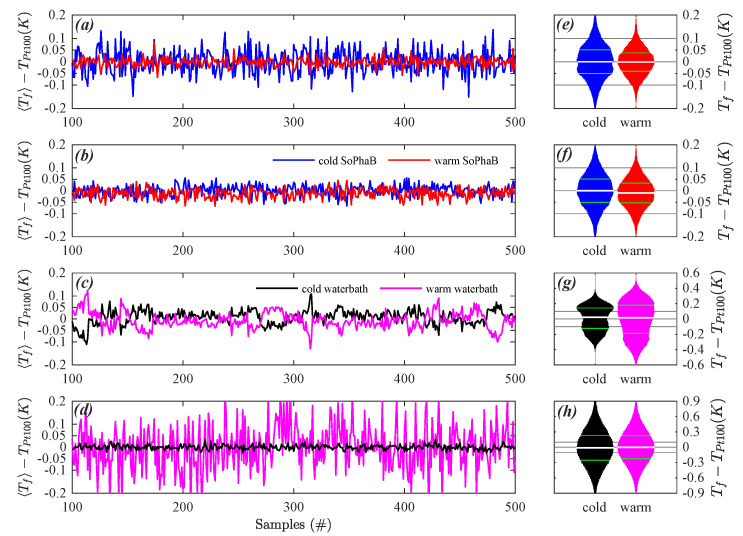
Solid-Phase Bath (SoPhaB) and water bath performance from the constant-temperature field evaluations: Temporal course of the differences between the spatially averaged fiber-optic and reference (Pt100) temperatures for a randomly selected subset of 500 samples (**a**–**d**), and probability density functions of the their instantaneous deviations for the entire cold and warm reference section (**e**–**h**) for (**a**,**e**) Arctic polarnight, (**b**,**f**) Urban summer, (**c**,**g**) Arctic pier, and (**d**,**h**) Grass summer. In violin subplots (**e**–**h**), the surface areas were normalized by the number of samples for each experiment. Horizontal grey lines are provided for zero and ±0.1 K for reference. The interquartile range is marked by the 25% and 75% percentiles (green short line), and the median (white short line).

**Table 1 sensors-22-04244-t001:** Short description of fiber-optic distributed sensing experiments and calibration parameters used for evaluating SoPhaB performance. SMM refers to single-core multimode glass fiber of 50 μm outer diameter (OD), avg for averaging, PVC is polyvinyl chloride, PE polyethylen, Pt100 the platinum wire reference thermometers, DTS is Distributed Temperature Sensing. FODS observations were selected for a 300 min period each. Names of calibration routines refer to methods described in des Tombe et al. [16] except for Windtunnel. Models of DTS instruments refer to those of Silixa Ltd., Hertfordshire, UK.

Reference Label	Fiber-Optic Cable	Fiber-Optic Sampling and Cable Length	DTS Instrument	Calibration Routine	Reference Bath
Windtunnel	SMM, stainless-steel sheath (0.82 mm OD), loosely buffered, gel-filled, with red PE coating (0.15 mm), OD 1.12 mm	0.127 m, 30 s avg sampled every 90 s, length 270 m	Silixa Ultima 5 km variant	internal manufacturer calibration with differential attenuation estimated visually; single-ended	SoPhaB, Pt100 read by external data logger
Arctic polarnight	SMM, with transparent PE jacket, OD 0.25 mm	0.254 m, 5 s avg sampled every 15 s, length 250 m	Silixa XT	ordinary least squares; single-ended	SoPhaB, Pt100 read by DTS machine
Urban summer	SMM, PVC sheath, tightly buffered, aramid reinforced, white, OD 0.9 mm	0.254 m, 12 s avg sampled every 12 s, length 610 m	Silixa XT	weighted least squares; double-ended	SoPhaB, Pt100 read by DTS machine
Arctic pier	SMM, PVC sheath, tightly buffered, aramid reinforced, white, OD 0.9 mm	0.254 m, 10 s avg sampled every 10 s, length 320 m	Silixa XT	ordinary least squares; single-ended	liquid-phase filled with antifreeze, crushed ice (cold), Pt100 read by DTS machine
Grass summer	SMM, PVC sheath, tightly buffered, aramid reinforced, white, OD 0.9 mm	0.125 m, 1 s avg sampled every 4 s, length 600 m	Silixa Ultima 2 km variant	ordinary least squares; single-ended	liquid-phase water, crushed ice (cold), Pt100 read by external data logger

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
