# Peer review of "Solid-Phase Reference Baths for Fiber-Optic Distributed Sensing"

_sensors, 2022, doi:10.3390/s22114244_

Round 1
Reviewer 1 Report
The authors propose and demonstrate the solid-phase as the new reference baths for fiber-optic distributed sensing. The use of solid-phase baths for referencing fiber-optic distributed sensing measurements offers many advantages over traditional liquid-phased baths. The experimental measurements realize the goal of improving the calibration and signal quality. Therefore, the design proposed in this paper can boost the application of optical fiber distributed sensing in the fields of geosciences, engineering, and industry, expanding the possibilities for devices developments. I believe this work is well-suited for publication in Sensors after minor revisions.
Minor issues:
- Line 164-165: This sentence “Assuming an absence of turbulent diffusion, here termed convection, in the cut-out groove of the inner core, the dominant heat transport mechanism to the fiber, which may or may not physically touch the inner core depending on how tightly it can be wound, is radiative transfer due to conduction in air being very small.” Here, whether the physically touch affects the results of the experiment? When the optical fiber is tightly attached to the device, whether the optical fiber strain effect can influence the device performance?
- The authors estimate the response time of device from Equation 1, can the authors provide either numerical simulations or experimental results to certify this results?
- Page 12: The caption of Fig.5. This sentence “The vertical dotted lines in 4a-d mark the beginning and end of the reference section over which the spatial average was computed.” The position of this sentence might be in Fig. 4.

Author Response
Response to comments from Reviewer 1 (anonymous), pleas note that original comments are preceded by R1 and numbered (Cx), followed by our response (Response Author RAx) in italics, and the revised text passages with updated line numbers:
R1: The authors propose and demonstrate the solid-phase as the new reference baths for fiber-optic distributed sensing. The use of solid-phase baths for referencing fiber-optic distributed sensing measurements offers many advantages over traditional liquid-phased baths. The experimental measurements realize the goal of improving the calibration and signal quality. Therefore, the design proposed in this paper can boost the application of optical fiber distributed sensing in the fields of geosciences, engineering, and industry, expanding the possibilities for devices developments. I believe this work is well-suited for publication in Sensors after minor revisions.
Response Authors (RA1): Thank you for your positive supportive assessment of our work with regard to utility and correctness of methods.
Minor issues:
- R1-C1: Line 164-165: This sentence “Assuming an absence of turbulent diffusion, here termed convection, in the cut-out groove of the inner core, the dominant heat transport mechanism to the fiber, which may or may not physically touch the inner core depending on how tightly it can be wound, is radiative transfer due to conduction in air being very small.” Here, whether the physically touch affects the results of the experiment? When the optical fiber is tightly attached to the device, whether the optical fiber strain effect can influence the device performance?
R1-RA1: You are correct, whether the fiber-optic cable physically touches the copper block is irrelevant given the overall time response and spatiotemporal variability of its body temperature. Since the fiber is not ’locked in a tightly wrapped state’ around the inner core, it will detach itself anyway to some degree. The statement in question was mainly meant to provide instructions for how to easily deploy the SoPhab: if the fiber-optic cable is not tightly wrapped around the inner core, closing the SoPhab by sliding the outer ring into place is difficult. We clarified this in the manuscript by editing the paragraph. It now reads (ln 173-180):
"Note that the response time and spatiotemporal variability of the readings from the fiber-optic cable wound around the inner core is independent of direct physical contact, as it is dominated by the time scale given in Eq. 1. Tightly wrapping the fiber-optic cable around the inner core helps sliding the outer ring into place when closing the SoPhab. However, applying too much tension will induce unwanted strain and result in changes of the optical light conducting properties, which may lead to step changes in the differential attenuation of the Raman intensities."
- R1-C2: The authors estimate the response time of device from Equation 1, can the authors provide either numerical simulations or experimental results to certify this results?
R1-RA2: Yes, we did verify the validity of the presented equations in a series of laboratory experiments using ultrafine copper slabs of different thickness. The experimental results agreed remarkably well with the predictions within 1 to 2 s. We added a text passage to the manuscript describing this experimental finding. It now reads (ln 149-152):
"The validity of Equation 1 to predict a meaningful time response was confirmed by a series of laboratory tests using slabs of ultrafine copper of varying thickness. The experimental findings agreed with their predictions within seconds (not shown here)."
- R1-C3: Page 12: The caption of Fig.5. This sentence “The vertical dotted lines in 4a-d mark the beginning and end of the reference section over which the spatial average was computed.” The position of this sentence might be in Fig. 4.
R1-RA3: Thank you for catching this oversight, we deleted this sentence from the caption of Fig. 5.
Reviewer 2 Report
In this paper, the authors present the design and the results from laboratory and field evaluations of a novel Solid-Phase Bath, short SoPhaB, using ultrafine copper instead of the traditional mechanically stirred liquid-phase water bath. This type is suitable for all FODS applications in geosciences and industry when high accuracy and precision are needed. This article is clear, concise, and suitable for the scope of the journal. Several small suggestions are supplied:
1. Suggest the authors give more detail about the Solid Phase Bath (SoPhaB) and water bath performance from the constant-temperature field evaluations.
2. Suggest the authors improve the introduction part, also add some last references on the same topic. Some latest references are suggested:
Optical fiber sensing for marine environment and marine structural health monitoring: A review Optics and Laser Technology, 2021.
Overview of Fibre Optic Sensing Technology in the Field of Physical Ocean Observation, Front. Phys., 2021
Author Response
Response to comments from Reviewer 2 (anonymous), pleas note that original comments are preceded by R2 and numbered (Cx), followed by our response (Response Author RAx) in italics, and the revised text passages with updated line numbers:
R2: In this paper, the authors present the design and the results from laboratory and field evaluations of a novel Solid-Phase Bath, short SoPhaB, using ultrafine copper instead of the traditional mechanically stirred liquid-phase water bath. This type is suitable for all FODS applications in geosciences and industry when high accuracy and precision are needed. This article is clear, concise, and suitable for the scope of the journal. Several small suggestions are supplied:
Response Authors (RA2): Thank you for your positive supportive assessment of our work with regard to utility and correctness of methods, and fit of scope for Sensors.
- R2-C1: Suggest the authors give more detail about the Solid Phase Bath (SoPhaB) and water bath performance from the constant-temperature field evaluations.
R2-RA1: The in-depth presentation of the results for the constant-temperature field evaluations of both the SoPhabs and traditional water baths are the main focus of the results section, and include experiment 2 to 5 described in Table 1. Their descriptions take up the bulk parts of the results and discussions sections. As a response to your comment, we clarified that in principal all these experiments should have been constant-temperature evaluations, with small imperfections given by the temperature controls for heating/cooling and the changes in environmental forcings. Given the very small changes in the thermally-controlled bath temperatures (e.g. Fig. 4 e-h, and Fig. 5 a-d), we believe this assumption is valid. We are happy to address more specific comments if they are provided to us. It now reads (ln 204-206):
"All experiments with the exception of the intentional instationary Windtunnel data represent constant-temperature field evaluations, which are discussed in depth in sections 3 and 4."
- R2-C2: Suggest the authors improve the introduction part, also add some last references on the same topic. Some latest references are suggested:
Optical fiber sensing for marine environment and marine structural health monitoring: A review. Optics and Laser Technology, 2021.
Overview of Fibre Optic Sensing Technology in the Field of Physical Ocean Observation, Front. Phys., 2021
R2-RA2: Thank you for bringing these valuable references to our attention, we added them to our introductory section. It now reads (ln 26-41, ln 551-554):
"Fiber-optic distributed sensing (FODS) applications in geo- and marine sciences, engineering, and industry have gained importance over the past decade, and now include the field of meteorology, hydrology, geology, oceanography and many industrial applications. The spatially explicit nature of the in-situ collected observations via installed fiber-optic cables allows for capturing unprecedented spatial scales at fine resolution across several orders of magnitude (from mm to km) at fine temporal resolution (from s to hours) depending on the geometry of the sensing cable(s) network. The latter may include vertical profiles [1–3 ], horizontal transects [4 –6], and 2-D [ 7] and quasi or full 3-D mesh grids [ 8 ,9 ] to name a few. The list of derived environmental quantities in the fields of meteorology and surface hydrology include temperature, flow velocity and direction, turbulence kinetic energy and thermal dissipation, solar and terrestrial radiation, salinity, air humidity, and soil moisture content to name a few. The reader is referred to Thomas and Selker [10] for a list of applications and techniques in these fields. Fiber-optic point and distributed sensing is also evolving rapidly in the neighboring marine sciences, with observable quantities including temperature, salinity, pressure, chemical and structural parameters, velocity and turbulence [11 , 12]
- Wang, L.; jie Wang, Y.; Song, S.; Li, F. Overview of Fibre Optic Sensing Technology in the Field of Physical Ocean Observation. Frontiers in Physics 2021, 9. doi:10.3389/fphy.2021.745487.
12. Min, R.; Liu, Z.; Pereira, L.; Yang, C.; Sui, Q.; Marques, C. Optical fiber sensing for marine environment and marine structural health monitoring: A review. Optics & Laser Technology 2021, 140, 107082. doi:10.1016/j.optlastec.2021.107082."